# Multigene phylogeny and taxonomy of *Dendryphion hydei* and *Torula hydei* spp. nov. from herbaceous litter in northern Thailand

**Junfu Li**[1,2,3,4], **Rajesh Jeewon**[5], **Peter E. Mortimer**[3], **Mingkwan Doilom**[3,6], **Rungtiwa Phookamsak**[3,4,6]*, **Itthayakorn Promputtha**[1,2]*

**1** Department of Biology, Faculty of Science, Chiang Mai University, Chiang Mai, Thailand, **2** Center of Excellence in Bioresources for Agriculture, Industry and Medicine, Faculty of Science, Chiang Mai University, Chiang Mai, Thailand, **3** Key Laboratory of Biodiversity and Biogeography, Chinese Academy of Sciences, Kunming Institute of Botany, Kunming, P.R. China, **4** Center of Excellence in Fungal Research, Mae Fah Luang University, Chiang Rai, Thailand, **5** Department of Health Sciences, Faculty of Science, University of Mauritius, Reduit, Mauritius, **6** World Agroforestry Centre, East and Central Asia, Kunming, P.R. China

* itthayakorn.p@cmu.ac.th (IP); jomjam.rp2@gmail.com (RP)

**Data Availability Statement:** All relevant data are within the paper.

## Abstract

During our studies on asexual fungi colonizing herbaceous litter in northern Thailand, we discovered two new fungal species, viz. *Dendryphion hydei* and *Torula hydei* spp. nov. The latter are examined, and their morphological characters are described as well as their DNA sequences from ribosomal and protein coding genes are analysed to infer their phylogenetic relationships with extant fungi. *Torula hydei* is different from other similar *Torula* species in having tiny and catenate conidia. *Dendryphion hydei* can be distinguished from other similar *Dendryphion* species in having large conidiophores and subhyaline to pale olivaceous brown, 2–4(–5)-septate conidia. Multigene phylogenetic analyses of a combined LSU, SSU, TEF1-α, RPB2 and ITS DNA sequence dataset generated from maximum likelihood and Bayesian inference analyses indicate that *T. hydei* forms a distinct lineage and basal to *T. fici*. *Dendryphion hydei* forms a distinct lineage and basal to *D. europaeum*, *D. comosum*, *D. aquaticum* and *D. fluminicola* within Torulaceae (Pleosporales, Dothideomycetes).

## Introduction

The family Torulaceae Corda was introduced by Sturm [1] and is typified by *Torula* Pers. Species in Torulaceae are known only by their asexual morphs which are characterized as followed: superficial, effuse, greyish brown to black, powdery colonies; micro- or macronematous conidiophores, with or without apical branches; doliiform to ellipsoid or clavate, brown, smooth to verruculose, mono- to polyblastic conidiogenous cells which often remaining cupulate; subcylindrical, phragmosporous, acrogenous, brown, dry, smooth to verrucose conidia characteristically produced in branched chains [2,3,4,5,6,7]. Crous et al. [8] investigated phylogenetic relationships of this family with the inclusion of *Torula* species and accepted *Dendryphion* Wallr., besides *Torula* within Torulaceae in Pleosporales. Su et al. [6] introduced *Neotorula* Ariyaw., Z.L. Luo & K.D. Hyde and two new *Dendryphion* species in Torulaceae

**Funding:** The authors are grateful the Mushroom Research Foundation, Chiang Mai, Thailand and the Key Research Program of Frontier Sciences of the Chinese Academy of Sciences (grant no. QYZDY-SSW-SMC014) for supporting this research. Rungtiwa Phookamsak thanks CAS President's International Fellowship Initiative (PIFI) for young staff (grant no. Y9215811Q1) and National Science Foundation of China (NSFC) project code 31850410489 (grant no. Y81I982211) for financial support. Peter E Mortimer would like to thank the National Science Foundation of China and the South East Asian Biodiversity resources Institute, Chinese Academy of Sciences, for financial support under the following grants: 41761144055, 41771063, Y4ZK111B01. Mingkwan Doilom thanks the 5th batch of Postdoctoral Orientation Training Personnel in Yunnan Province and the 64th batch of China Postdoctoral Science Foundation. The funders had no role in study design, data collection and analysis, decision to publish, or preparation of the manuscript.

**Competing interests:** NO authors have competing interests.

based on molecular data. Li et al. [9] established a novel genus, *Sporidesmioides* Jun F. Li, Phook. & K.D. Hyde. Su et al. [7] examined 21 freshwater taxa and updated phylogenetic relationships of taxa within the family Torulaceae based on ITS, LSU, TEF1-α and RPB2 genes and accommodated *Rostriconidium* Z.L. Luo, K.D. Hyde & H.Y. Su within Torulaceae. Crous et al. [10] designated the epitype of *Rutola* J.L. Crane & Schokn. and accepted the genus in Torulaceae based on LSU phylogeny. Currently, there are six accepted genera in Torulaceae viz. *Dendryphion*, *Neotorula*, *Rostriconidium*, *Rutola*, *Sporidesmioides* and *Torula* [10,4,9,6,7].

*Torula* is typified by *T. herbarum* Pers. and is morphologically characterized by having terminal or lateral, monoblastic or polyblastic conidiogenous cells with a thickened and heavily melanized wall on the base and thin-walled and frequently collapsing and becoming coronate on the apex [11]. Crane and Schoknecht [12] provided details of conidiogenesis in *Torula* based on light and transmission electron microscopy. Based on their examination, conidiogenesis has provided good taxonomic insights useful to segregate *Torula* and these were also observed by Mason [13], Hughes [14], Subramanian [15] and Ellis [16,17]. However, there was little information regarding the phylogenetic relationships of *Torula* until the studies of Crous et al. [8], Li et al. [5] and Su et al. [6,7]. To date, only 15 species have their DNA sequence data being analysed to reveal their phylogenetic placements in Torulaceae [18,19,9,5,6,7,20].

*Dendryphion* Wallr. was introduced by Wallroth [21] to accommodate hyphomycetous species, *D. comosum* Wallr. The genus is commonly known to be saprobic on dead stems of herbaceous plants and decaying wood, and is characterized by having erect, solitary, branched in upper part, polytretic conidiophores, forming septate, pigmented, thick-walled, finely roughened stipe and a distinct conidiogenous apparatus, with dark scars and catenate, in simple or branched chains of brown, septate (didymo- or cheiro) conidia [8,7]. Crous et al. [3] introduced *D. europaeum* Crous & R.K. Schumacher based on morphological characteristics and molecular data and later Crous et al. [8] accommodated the species in Torulaceae and further accepted *Dendryphion* in Torulaceae. Su et al. [6] circumscribed genera of Torulaceae from freshwater. Only seven *Dendryphion* species have DNA sequence data and their phylogenetic affinities to members of the Torulaceae have been investigated.

In this study, a novel *Torula* species was isolated from herbaceous litters collected from northern Thailand. Among collected samples, *Dendryphion hydei* is also recovered as another new species from northern Thailand. These species are described and illustrated. In addition, an updated phylogenetic tree with our new taxa for the family Torulaceae is provided in this study.

## Material and methods

### Isolation and identification

The specimens were collected from herbaceous litters (*Chromolaena odorata* Linn. and *Bidens pilosa* Linn.) in northern Thailand during the year 2015 to 2016. Samples were returned to the laboratory (Center of Excellence in Fungal Research, Mae Fah Luang University, Chiang Rai, Thailand) for examination and description of morphological characteristics. The specimens were observed under a Motic SMZ 168 series dissecting stereomicroscope. The conidial structures were picked up by a sterilized surgical needle and transferred into 10% lacto-glycerol on a clean slide and examined under a Nikon Eclipse 80i compound microscope and photo-captured with a Canon 600D digital camera using DIC microscopy. Macro- morphological structures were photographed with a Discovery V.8 stereo microscope fitted with a CARL ZEISS Axio Cam ERc5S microscope camera. Tarosoft® Image Frame Work program v.0.9.0.7 and Adobe Photoshop CS5 Extended version 10.0 software (Adobe Systems Inc., The United States) were used for measurements and drawing photographic plates.

Single conidium isolation was carried out to obtain pure cultures as described in Dai et al. [22]. Germinating conidia were transferred aseptically to potato dextrose agar (PDA) and malt extract agar (MEA) plates and grown at room temperature (16–30°C) in alternating day and night light. Colony characters were observed and recorded after one week and at weekly intervals [23,24].

The type specimens were deposited in the herbarium of Mae Fah Luang University (MFLU), Chiang Rai, Thailand and the Herbarium of Cryptogams Kunming Institute of Botany Academia Sinica (KUN-HKAS), Yunnan, China. Ex-type living cultures were deposited in Mae Fah Luang University Culture Collection (MFLUCC 18–0250 and MFUCC 18–0236) and Kunming Institute of Botany Culture Collection (KUMCC 16–0037 and KUMCC 18–0009). Faces of Fungi and Index Fungorum numbers are registered as outlined in Jayasiri et al. [25] and Index Fungorum [26]. New species are established based on guidelines of Jeewon and Hyde [27].

## DNA extraction, PCR amplification and sequencing

Fungal mycelium was scraped off and transferred to a 1.5 ml micro-centrifuge tube using a sterilized lancet for genomic DNA extraction. The Biospin Fungus Genomic DNA Extraction Kit-BSC14S1 (BioFlux®, P.R. China) was used to extract fungal genomic DNA, following the protocols in the manufacturer's instructions.

DNA amplification was performed by polymerase chain reaction (PCR) using the following genes (ITS, LSU, SSU, RPB2 and TEF1-α). The primers ITS5 and ITS4 primer pairs were used to amplify the ITS and 5.8S regions of the rDNA gene [28]; The primers LR0R and LR5 were used to amplify the partial ribosomal RNA for the 28S nuclear large subunit (LSU) [29]; NS1 and NS4 were used to amplify the partial ribosomal RNA for the 18S nuclear small subunit (SSU) [28]; fRPB2-5F and fRPB2-7cR were used to amplify the partial RNA polymerase second largest subunit (RPB2) [30] and EF1-983F and EF1-2218R were used to amplify the translation elongation factor 1-alpha gene (TEF1-α) [31].

The final volume of the PCR reaction was 25 μl, containing 1 μl of DNA template, 1 μl of each forward and reward primer, 12.5 μl of 2×Easy Taq PCR SuperMix (mixture of *Easy-Taq*TM DNA Polymerase, dNTPs, and optimized buffer, Beijing TransGen Biotech Co., Ltd., Beijing, P.R. China) and 9.5 μl of ddH$_2$O. The PCR thermal cycling conditions of ITS, LSU, SSU and TEF1-α were as follows: 94°C for 3 minutes, followed by 35 cycles of denaturation at 94°C for 30 seconds, annealing at 55°C for 50 seconds, elongation at 72°C for 1 minute, and a final extension at 72°C for 10 minutes. The PCR thermal cycle program for RPB2 was as follows: initial denaturation at 95°C for 5 minutes, followed by 40 cycles of denaturation at 95°C for 1 minute, annealing at 52°C for 2 minutes, elongation at 72°C for 90 seconds, and final extension at 72°C for 10 minutes. Purification and sequencing of PCR fragments with PCR primers mentioned above were carried out at Shanghai Majorbio Biopharm Technology Co., Ltd, China.

## Sequence alignment and phylogenetic analyses

Phylogenetic analyses were performed from single gene (LSU dataset) as well as based on a combined LSU, SSU, TEF1-α, RPB2 and ITS sequence dataset. Sequences generated from this study were analyzed with other similar sequences obtained from GenBank and those derived from recent publications [32,10,19,9,5,6,7] (Table 1). The single gene alignment was performed by using MAFFT v. 7 [33] (http://mafft.cbrc.jp/alignment/server/) and manually aligned wherever necessary in MEGA version 7.0 [34]. Further analyses for the combined dataset were

**Table 1. Taxa used in the phylogenetic analysis and their corresponding GenBank numbers.** The newly generated sequences are indicated in **blue bold** font, while the type strains are in **black bold** font.

| Species | Culture collection/ Voucher no. | GenBank accession numbers | | | | | References |
|---|---|---|---|---|---|---|---|
| | | ITS | LSU | SSU | RPB2 | TEF1-α | |
| *Arthopyrenia salicis* | CBS 368.94 | KF443410 | AY779288 | AY538333 | KF443397 | KF443404 | [41] |
| ***Cycasicola goaensis*** | **MFLUCC 17–0754** | **MG828885** | **MG829001** | **MG829112** | – | **MG829198** | [42] |
| ***Dendryphion aquaticum*** | **MFLUCC 15–0257** | **KU500566** | **KU500573** | **KU500580** | – | – | [6] |
| ***Dendryphion comosum*** | **CBS 208.69** | **MH859293** | **MH871026** | – | – | – | [43] |
| *Dendryphion europaeum* | CPC 22943 | KJ869146 | KJ869203 | – | – | – | [3] |
| ***Dendryphion europaeum*** | **CPC 23231** | **KJ869145** | **KJ869202** | – | – | – | |
| *Dendryphion fluminicola* | KUMCC 15–0321 | MG208160 | MG208139 | – | MG207971 | MG207990 | [7] |
| *Dendryphion fluminicola* | DLUCC 0849 | MG208161 | MG208140 | – | MG207972 | MG207991 | |
| ***Dendryphion fluminicola*** | **MFLUCC17-1689** | **NR_157490** | **MG208141** | – | – | **MG207992** | |
| *Dendryphion hydei* | KUMCC 18–0009 | **MN061343** | MH253927 | MH253929 | – | MH253931 | This study |
| *Dendryphion nanum* | HKAS84010 | KU500568 | KU500575 | KU500582 | – | – | [6] |
| *Dendryphion nanum* | HKAS84012 | KU500567 | KU500574 | KU500581 | – | – | |
| *Dendryphion nanum* | MFLUCC 16–0987 | MG208156 | MG208135 | – | MG207967 | MG207986 | [7] |
| ***Dendryphion submersum*** | **MFLUCC15-0271** | **KU500565** | **KU500572** | **KU500579** | – | – | [6] |
| *Dendryphion submersum* | KUMCC15-0455 | MG208159 | MG208138 | – | MG207970 | MG207989 | [7] |
| ***Hobus wogradensis*** | **CBS 141484** | **NR_147652** | **KX650546** | **NG_061253** | **KX650575** | **KX650521** | [44] |
| ***Liua muriformis*** | **KUMCC 18–0177** | **MK433599** | **MK433598** | **MK433595** | **MK426799** | **MK426798** | [45] |
| ***Neooccultibambusa chiangraiensis*** | **MFLUCC 12–0584** | **NR_154238** | **KU764699** | **KU712458** | – | – | [46] |
| ***Neoroussoella bambusae*** | **MFLUCC 11–0124** | **KJ474827** | **KJ474839** | – | **KJ474856** | **KJ474848** | [47] |
| ***Neotorula aquatica*** | **MFLUCC 15–0342** | **KU500569** | **KU500576** | **KU500583** | – | – | [6] |
| *Neotorula submersa* | HKAS 92660 | NR_154247 | KX789217 | – | – | – | [4] |
| *Nigrograna mackinnonii* | E5202H | JK26415 | KJ605422 | JK264155 | JK264156 | JK264154 | [48] |
| *Nigrograna mackinnonii* | CBS 110022 | KF015653 | KF015609 | GQ387553 | KF015704 | KF407985 | [41] |
| ***Nigrograna mackinnonii*** | **CBS 674.75** | **NR_132037** | **GQ387613** | **GQ387552** | – | – | |
| ***Nigrograna marina*** | **CY 1228** | – | **GQ925848** | **GQ925835** | **GU479823** | **GU479848** | [49] |
| ***Occultibambusa bambusae*** | **MFLUCC 13–0855** | **KU940123** | **KU863112** | **KU872116** | **KU940170** | **KU940193** | [22] |
| *Ohleria modesta* | WU 36870 | KX650562 | – | – | KX650582 | KX650533 | [44] |
| *Ohleria modesta* | CBS 141480 | KX650563 | – | KX650513 | KX650583 | KX650534 | |
| ***Parathyridaria ramulicola*** | **CBS 141479** | **NR_147657** | **KX650565** | **KX650514** | **KX650584** | **KX650536** | [44] |
| ***Parathyridaria percutanea*** | **CBS 868.95** | **NR_147631** | **NG_058022** | **NG_062999** | **KF366452** | **KF407987** | [41] |
| ***Parathyridaria robiniae*** | **MFLUCC 14–1119** | **KY511142** | **KY511141** | – | – | **KY549682** | [20] |
| ***Roussoella chiangraina*** | **MFLUCC 10–0556** | **NR_155712** | **KJ474840** | – | **KJ474857** | **KJ474849** | [47] |
| ***Roussoella nitidula*** | **MFLUCC 11–0182** | **KJ474835** | **KJ474843** | – | **KJ474859** | **KJ474852** | [47] |
| *Roussoella scabrispora* | MFLUCC 11–0624 | KJ474836 | KJ474844 | – | KJ474860 | KJ474853 | [47] |
| *Rostriconidium aquaticum* | KUMCC 15–0297 | MG208165 | MG208144 | – | MG207975 | MG207995 | [7] |
| ***Rostriconidium aquaticum*** | **MFLUCC 16–1113** | **MG208164** | **MG208143** | – | **MG207974** | **MG207994** | |
| ***Rostriconidium pandanicola*** | **KUMCC 17–0176** | **MH275084** | **MH260318** | **MH260358** | **MH412759** | **MH412781** | [50] |
| ***Roussoellopsis macrospora*** | **MFLUCC 12–0005** | **KJ739604** | **KJ474847** | **KJ739608** | **KJ474862** | **KJ474855** | [47] |
| *Roussoellopsis tosaensis* | KT1659 | – | AB524625 | AB524484 | AB539104 | AB539117 | [51] |
| ***Rutola graminis*** | **CPC 33267** | **MN313814** | **MN317295** | – | – | – | [10] |
| *Rutola graminis* | CPC 33695 | MN313815 | MN317296 | – | – | – | |
| *Rutola graminis* | CPC 33715 | MN313816 | MN317297 | – | – | – | |
| *Sporidesmium australiense* | HKUCC 10833 | – | DQ408554 | – | DQ435080 | – | [52] |
| ***Sporidesmioides thailandica*** | **MFLUCC 13–0840** | **MN061347** | **NG_059703** | **NG_061242** | **KX437761** | **KX437766** | [9] |
| *Sporidesmioides thailandica* | KUMCC 16–0012 | MN061348 | KX437758 | KX437760 | KX437762 | KX437767 | |

*(Continued)*

**Table 1.** (Continued)

| Species | Culture collection/ Voucher no. | GenBank accession numbers | | | | | References |
|---|---|---|---|---|---|---|---|
| | | ITS | LSU | SSU | RPB2 | TEF1-α | |
| *Thyridaria broussonetiae* | **CBS 141481** | **NR_147658** | **KX650568** | **NG_063067** | **KX650586** | **KX650539** | [44] |
| *Thyridaria broussonetiae* | CBS 121895 | KX650567 | KX650567 | – | KX650585 | KX650538 | |
| *Thyridariella mahakashae* | **NFCCl 4215** | **MG020435** | **MG020438** | **MG020441** | **MG020446** | **MG023140** | [53] |
| *Thyridariella mangrovei* | **NFCCl 4213** | **MG020434** | **MG020437** | **MG020440** | **MG020445** | **MG020443** | [53] |
| *Torula acaciae* | **CPC 29737** | **NR_155944** | **NG_059764** | – | **KY173594** | – | [54] |
| *Torula aquatica* | DLUCC 0550 | MG208166 | MG208145 | – | MG207976 | MG207996 | [7] |
| *Torula aquatica* | MFLUCC16-1115 | MG208167 | MG208146 | – | MG207977 | – | |
| *Torula breviconidiophora* | **KUMCC 18–0130** | **MK071670** | **MK071672** | **MK071697** | – | **MK077673** | [19] |
| *Torula camporesii* | **KUMCC 19–0112** | **MN507400** | **MN507402** | **MN507401** | **MN507404** | **MN507403** | [55] |
| *Torula chiangmaiensis* | **KUMCC 16–0039** | **MN061342** | **KY197856** | **KY197863** | – | **KY197876** | [5] |
| *Torula chromolaenae* | **KUMCC 16–0036** | **MN061345** | **KY197860** | **KY197867** | **KY197873** | **KY197880** | [5] |
| *Torula fici* | **CBS 595.96** | **KF443408** | **KF443385** | **KF443387** | **KF443395** | **KF443402** | [8] |
| *Torula fici* | KUMCC 15–0428 | MG208172 | MG208151 | – | MG207981 | MG207999 | [7] |
| *Torula fici* | KUMCC 16–0038 | MN061341 | KY197859 | KY197866 | KY197872 | KY197879 | [5] |
| *Torula gaodangensis* | **MFLUCC 17–0234** | **MF034135** | **NG_059827** | **NG_063641** | – | – | [18] |
| *Torula goaensis* | **NFCCL 4040** | **NR_159045** | **NG_060016** | – | – | – | [56] |
| *Torula herbarum* | **CPC 24414** | **KR873260** | **KR873288** | – | – | – | [8] |
| *Torula hollandica* | **CBS 220.69** | **NR_132893** | **NG_064274** | **KF443389** | **KF443393** | **KF443401** | [8] |
| *Torula hydei* | **KUMCC 16–0037** | **MN061346** | **MH253926** | **MH253928** | – | **MH253930** | This study |
| *Torula mackenziei* | **MFLUCC 13–0839** | **MN061344** | **KY197861** | **KY197868** | **KY197874** | **KY197881** | [5] |
| *Torula masonii* | **CBS 245.57** | **NR_145193** | **NG_058185** | – | – | – | [8] |
| *Torula masonii* | DLUCC 0588 | MG208173 | MG208152 | – | MG207982 | MG208000 | [6] |
| *Torula masonii* | KUMCC 16–0033 | MN061339 | KY197857 | KY197864 | KY197870 | KY197877 | [5] |
| *Torula pluriseptata* | **MFLUCC 14–0437** | **MN061338** | **KY197855** | **KY197862** | **KY197869** | **KY197875** | [5] |
| *Torula polyseptata* | **KUMCC 18–0131** | **MK071671** | **MK071673** | **MK071698** | – | **MK077674** | [19] |
| *Torula* sp. | CBS 246.57 | KF443411 | KR873290 | – | – | – | [8] |
| . | | | | | | | |

**Abbreviations: CBS**: Westerdijk Fungal Biodiversity Institute, Utrecht, The Netherlands; **CPC**: Collection of Pedro Crous housed at CBS; **DLUCC**: Dali University Culture Collecting Center, Dali, Yunnan, China. **HKAS**: Herbarium of Cryptogams Kunming Institute of Botany Academia Sinica (HKAS), Yunnan, China; **HKUCC**: University of Hong Kong Culture Collection, Department of Ecology and Biodiversity, Hong Kong, China; **KUMCC**: Kunming Institute of Botany Culture Collection, Chinese Science Academy, Kunming, China; **MFLUCC**: Mae Fah Luang University Culture Collection, Chiang Rai, Thailand; **NFCCI**: National Fungal Culture Collection of India; **KT**: K. Tanaka.

analyzed by maximum likelihood (ML) implemented in RAxMLGUI v.0.9b2 [35,36,37,38] and Bayesian Inference (BI) criteria [39,40] following the methodology in Li et al. [5].

The phylogram was represented in Treeview [57] and drawn in Microsoft PowerPoint and converted to jpeg file in Adobe Photoshop version CS5 (Adobe Systems Inc., the United States). The new sequences were submitted in GenBank (Table 1). The alignment was deposited in TreeBASE [58] under the accession number 25462.

## Nomenclature

The electronic version of this article in Portable Document Format (PDF) in a work with an ISSN or ISBN will represent a published work according to the International Code of Nomenclature for algae, fungi, and plants, and hence the new names contained in the electronic

publication of a PLOS ONE article are effectively published under that Code from the electronic edition alone, so there is no longer any need to provide printed copies.

In addition, new names contained in this work have been submitted to Index Fungorum from where they will be made available to the Global Names Index. The unique Index Fungorum number can be resolved and the associated information viewed through any standard web browser by appending the Index Fungorum number contained in this publication to the prefix www.indexfungorum.org/. The online version of this work is archived and available from the following digital repositories: PubMed Central and LOCKSS.

## Compliance with ethical standards

There is no conflict of interest (financial or non-financial) and all authors have agreed to submission of paper. The authors also declare that they have no conflict of interest and confirm that the field studies did not involve endangered or protected species.

## Results

### Phylogenetic analyses

The combined LSU, SSU, TEF1-α, RPB2 and ITS sequence dataset comprises 71 taxa with *Occultibambusa bambusae* (MFLUCC 13–0855) and *Neooccultibambusa chiangraiensis* (MFLUCC 12–0559) as the outgroup taxa. Bayesian Inference (BI) and maximum likelihood (ML) analyses of the combined dataset were performed to determine the placement of our new taxa and infer relationships at the intrageneric level as well as resolving the phylogenetic relationships of the core families in Pleosporales. The phylogenetic trees obtained from BI and ML analyses resulted in trees with largely similar topologies and also similar to those generated from previous studies based on maximum likelihood analysis [18,5,7]. The best scoring RAxML tree is shown in Fig 1, with the final ML optimization likelihood value of -32357.09 0382 (ln). The dataset consists of 4053 total characters including gaps (LSU: 1–840 bp, SSU: 841–1776 bp, TEF1-α: 1777–2566 bp, RPB2: 2567–3418 bp, ITS: 3419–4053). RAxML analysis yielded 1585 distinct alignment patterns and 33.97% of undetermined characters or gaps. Estimated base frequencies were as follows: A = 0.246366, C = 0.258260, G = 0.271248, T = 0.2 24126, with substitution rates AC = 1.424215, AG = 3.485957, AT = 1.457990, CG = 0.955364, CT = 6.607514, GT = 1.000000. The proportion of invariable sites I = 0, the gamma distribution shape parameter alpha = 0.180234 and the Tree-Length = 3.299994. Bayesian posterior probabilities (BYPP) from MCMC were evaluated with final average standard deviation of split frequencies = 0.008574.

Most of the core genera of Torulaceae and other representative genera in Nigrogranaceae, Ohleriaceae, Roussoellaceae and Thyridariaceae are included in our phylogenetic analysis (Fig 1). Torulaceae formed a well-resolved clade (100% ML and 1.00 PP) with a close relationship to Roussoellaceae and Thyridariaceae. Species of different genera currently accommodated in Torulaceae formed well-resolved subclades except for *Sporidesmioides* which is recovered as basal to other genera with significant Bayesian support (1.00 PP) but with low support in ML analysis (56% ML). *Torula* is recovered as a strongly monophyletic genus in Torulaceae. *Torula hydei* is sister to *T. fici* with high support (100% ML and 1.00 PP). *Dendryphion hydei* forms a distinct lineage and related to *D. europaeum*, *D. comosum*, *D. aquaticum*, *D. fluminicola* and *D. submersum* with significant support in BI analysis (1.00 PP).

### Taxonomy

***Dendryphion hydei*** J.F. Li, Phookamsak & Jeewon, *sp. nov.* Fig 2

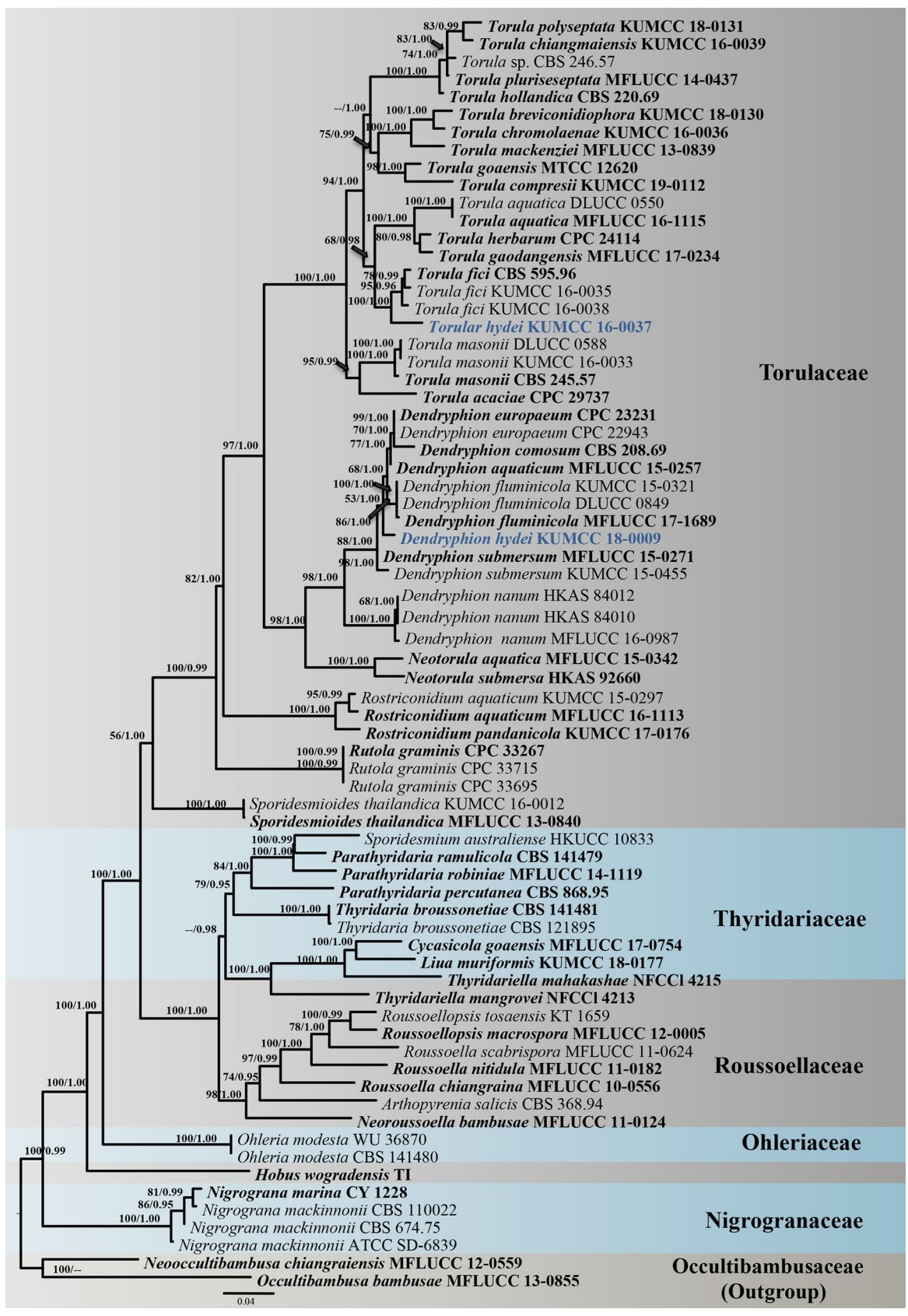

**Fig 1. Phylogenetic construction using RAxML-based analysis of a combined LSU, SSU, TEF1-α, RPB2 and ITS DNA sequence dataset.** Bootstrap support values for maximum likelihood (ML) equal to or greater than 50% and Bayesian posterior probabilities (PP) equal to or greater than 0.95 are shown as "ML/PP" above the nodes. The tree is rooted to *Occultibambusa bambusae* (MFLUCC 13–0855) and *Neooccultibambusa chiangraiensis* (MFLUCC 12–0559). The type strains are in black bold and the newly generated sequences are indicated in blue bold.

[urn:lsid:indexfungorum.org:names:556746]

Facesoffungi number: FoF04574

Etymology–Named in honour of Kevin D. Hyde for his excellent contribution to mycology and on his 65th birthday celebration.

Holotype–KUN-HKAS 97502

Saprobic on a branch litter of *Bidens pilosa* Linn. (Asteraceae). **Sexual morph**: Undetermined. **Asexual morph**: *Colonies* on the substratum superficial, effuse, gregarious, hairy, brown to dark brown. *Mycelium* composed of branched, septate, pale brown to brown hyphae. *Conidiophores* 260–380 μm long × 7–14 μm diam. (13–17 μm diam. at the base) ($\bar{x}$ = 356.7 × 9.9 μm, n = 10) macronematous, mononematous, septate, verrucose, thick-walled, branching simple or penicillate at the tip of primary branches, brown, flexuous. *Conidiogenous cells* 6–10 μm long × 3–5 μm diam. ($\bar{x}$ = 8 × 3.8 μm, n = 20) terminal, integrated, pale brown, polytretic. *Conidia* (17–)20–30(–35) μm long × 4–7 μm diam. ($\bar{x}$ = 26.5 μ 5.6 μm, n = 30) single, subhyaline to pale olivaceous brown, slightly paler at the end cells, dry, verrucose, monilioid, 2–4(–5)-septate, constricted at the septa. Conidial secession schizolytic.

Cultural characteristics: Conidia germinating on PDA within 14 hours and germ tubes produced from the apex. Colonies growing on PDA, reaching 5 cm in 21 days at 16–30°C, mycelium partly superficial, partly immersed, slightly effuse, hairy, vertical, with regular edge, white to grayish-brown, not produced pigmentation on media agar.

Material examined: THAILAND, Chiang Mai Province, Mae Taeng District, Mushroom Research Centre, on a branch litter of *Bidens pilosa* Linn., 12 July 2016, J.F. Li, FHP3 (HKAS 97502, **holotype**), ex-type living culture, MFLUCC 18–0236, KUMCC 18–0009.

Notes–*Dendryphion hydei* is unique in having large conidiophores and subhyaline to pale olivaceous brown, 2–4(–5)-septate conidia to compare with other related species in *Dendryphion*. *Dendryphion hydei* resembles *D. aquaticum* and *D. europaeum* in morphology. However, these species can be distinguished based on the size of the conidiophores, conidiogenous cells and conidia, as well as conidial septation and habitats (see Table 2). *Dendryphion hydei* has 2–4(–5)-septate conidia and inhabit in a terrestrial environment, similar to *D. europaeum*. However, *D. europaeum* has smaller conidiophores and conidia, and the conidia of *D. europaeum* are (2–)3(–5)-septate while *D. aquaticum* inhabits in a freshwater environment and has 3–6-septate conidia [3,7]. In the phylogenetic tree, *D. hydei* forms a separate lineage and clustered with *D. europaeum*, *D. comosum*, *D. aquaticum* and *D. fluminicola* with significant support in Bayesian inference analysis (1.00 PP). A comparison of TEF1-α nucleotides shows that *D. hydei* differs from *D. fluminicola* in 20/852 bp (2.3% difference, no gap) and from *D. submersum* in 30/902 bp (3.3% difference, no gap). A comparison of ITS nucleotides shows that *D. hydei* differs from *D. europaeum* in 19/553 bp (3.4% difference, no gap) and differs from *D. aquaticum* in 6/398 bp (1.5% difference, no gap). Phylogenetic analyses support *D. hydei* as a new species in *Dendryphion*. These tally with recommendations outlined by Jeewon and Hyde [27] to establish our new species. In this study, we collected *D. hydei* from *Bidens pilosa*, which is a new host record for this species. A morphometric comparison of the new taxon with other similar taxa of *Dendryphion* provide in Table 2.

***Torula hydei*** J.F. Li, Phookamsak & Jeewon, *sp. nov*. Fig 3

[urn:lsid:indexfungorum.org:names:556747]

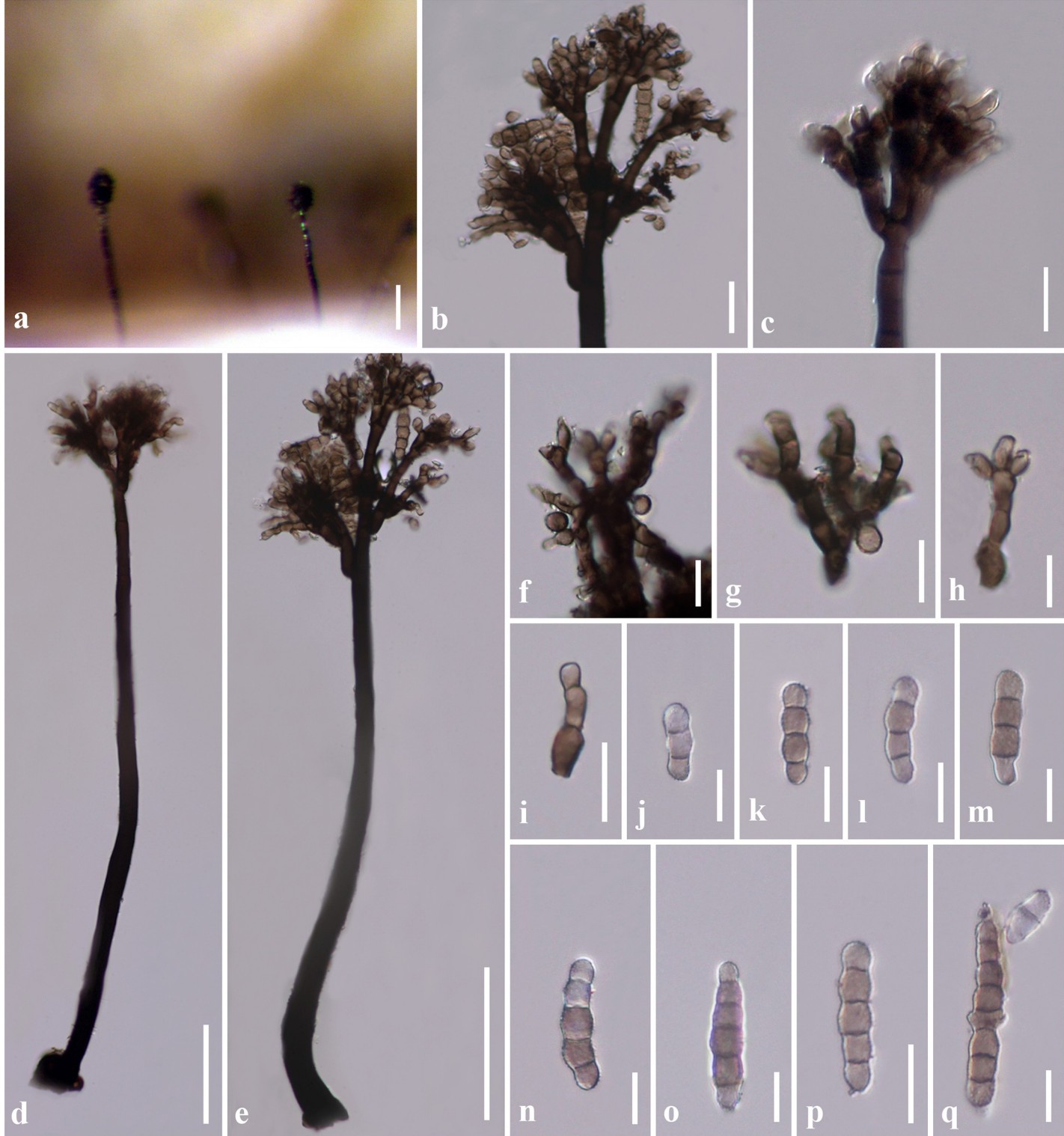

**Fig 2.** *Dendryphion hydei* (HKAS 97479, holotype) a Colonies on branch of *Bidens pilosa*. b, c Apex of conidiophores with conidial structures. d, e Conidiophores. f–i Conidiogenous cells. j–q Conidia. Scale bars: a = 100 μm, d, e = 50 μm, b, f–i = 20 μm, b, c, f–q = 10 μm.

**Table 2. Synopsis of morphological features of *Dendryphion* species discussed in this study.**

| Species | Size (μm) | | | Conidial septation | Host/substrate and habitat | Distribution | Reference |
|---|---|---|---|---|---|---|---|
| | Conidiophores | Conidiogenous cells | Conidia | | | | |
| *Dendryphion hydei* | 260–380 × 7–14 | 6–10 × 3–5 | (17–)20–30(–35) × 4–7 | 2–4(–5) | Branch litter of *Bidens pilosa* | Thailand | This study |
| *Dendryphion aquaticum* | 250–285 × 7.5–11.5 | 5–9 × 4–6 | 22–33 × 6.5–7.5 | 3–6 | Decaying wood submerged in stream | China (Yunnan) | [6] |
| *Dendryphion comosum* | Up to 400 × 9–14 | Up to16 × 5–8 | 9–65 × 5–9 | 1–5(–9) | Various hosts and substrates | Cosmopolitan distribution | [59, 60] |
| *Dendryphion europaeum* | 180–250 × 8–10 | 6–10 × 5–7 | (15–)20–28(–33) × (6–)7 | (2–)3(–5) | *Hedera helix*, *Heracleum sphondylium* | Germany, Netherlands | [3] |
| *Dendryphion fluminicola* | 114–176 × 7–10 | N/A | 31–46 × 8–9 | 2–6 | Decaying wood submerged in a stream in Cangshan Mountain, Lancang River and Jinsha River | China (Yunnan) | [7] |
| *Dendryphion nanum* | 52–64 × 6.5–8.5 | 13–19 × 6–8 | 56.7–74.5 × 10–12 | 3–11 | Various hosts and substrates | Cosmopolitan distribution | [59,6] |
| *Dendryphion submersum* | 210–335 × 3.5–4.5 | 11–15 × 4.5–6.5 | 15–25 × 5–7 | 2–5 | Decaying wood submerged in stream | China (Yunnan) | [6] |

Facesoffungi number: FoF 04573

Etymology–Named in honour of Kevin D. Hyde for his excellent contribution to mycology and on his 65th birthday celebration.

Holotype–HKAS 97478

Saprobic on an aerial dead branch of *Chromolaena odorata* Linn. **Sexual morph**: Undetermined. **Asexual morph**: *Colonies* discrete on host, black, powdery. *Mycelium* immersed on the substrate, composed of septate, branched, smooth, light brown hyphae. *Conidiophores* (1.5–)2–3 μm long × 1.5–2 μm diam. ($\bar{x}$ = 2.2 × 1.8 μm, n = 10), semi-macronematous, mononematous, solitary, erect, light brown, verruculose, thick-walled, consist of one cell or reduced to conidiogenous cells, without apical branches, subcylindrical to subglobose, arising from prostrate hyphae. *Conidiogenous cells* 3–5.5 μm long × 4.3–5 μm diam. ($\bar{x}$ = 3.8 × 4.5 μm, n = 20), polyblastic, terminal, dark brown to black, smooth to minutely verruculose, thick-walled, doliiform to ellipsoid. *Conidia* (7.5–)8–14 μm long × 2–4 μm diam. ($\bar{x}$ = 10.4 × 3.4 μm, n = 30), solitary to catenate, acrogenous, simple, phragmosporous, brown to dark brown, minutely verruculose, 2–3-septate, rounded at both ends, composed of subglobose cells, slightly constricted at some septa, chiefly subcylindrical. *Conidial secession* schizolytic.

Cultural characteristics: Conidia germinating on PDA within 14 hours and germ tubes produced from the apex. Colonies growing on PDA, reaching 5 cm in 10 days at 16–30°C, mycelium partly superficial, partly immersed, slightly effuse, hairy, vertical, with regular edge, light brown to brown, not produced pigmentation on media agar; not sporulated on media agar within 2 months.

Material examined: THAILAND, Chiang Mai Province, Mae Taeng District, on an aerial dead branch of *Chromolaena odorata* Linn. (Asteraceae), 26 December 2015, J.F. Li, MRC2 (HKAS 97478, **holotype**), ex-type living culture, MFLUCC 18–0250, KUMCC 16–0037.

Notes–*Torula hydei* resembles *T. herbarum* and *T. fici* in having 2–3-septate, catenated, brown, verruculose conidia, but differs in having smaller conidia [3]. Phylogenetic analyses showed that *T. hydei* constitutes an independent lineage basal to *T. fici* (100% ML and 1.00 BYPP). Morphologically *T. hydei* differs from *T. fici* in having smaller conidia (*T. hydei*, (7.5–)8–14 × 2–4 μm versus (12–)13–17(–19) × 5(–6) μm, *T. fici*) and the conidia are also brown to dark brown, paler at the apex where branching occurs [8]. Whereas, *T. fici* has brown conidia,

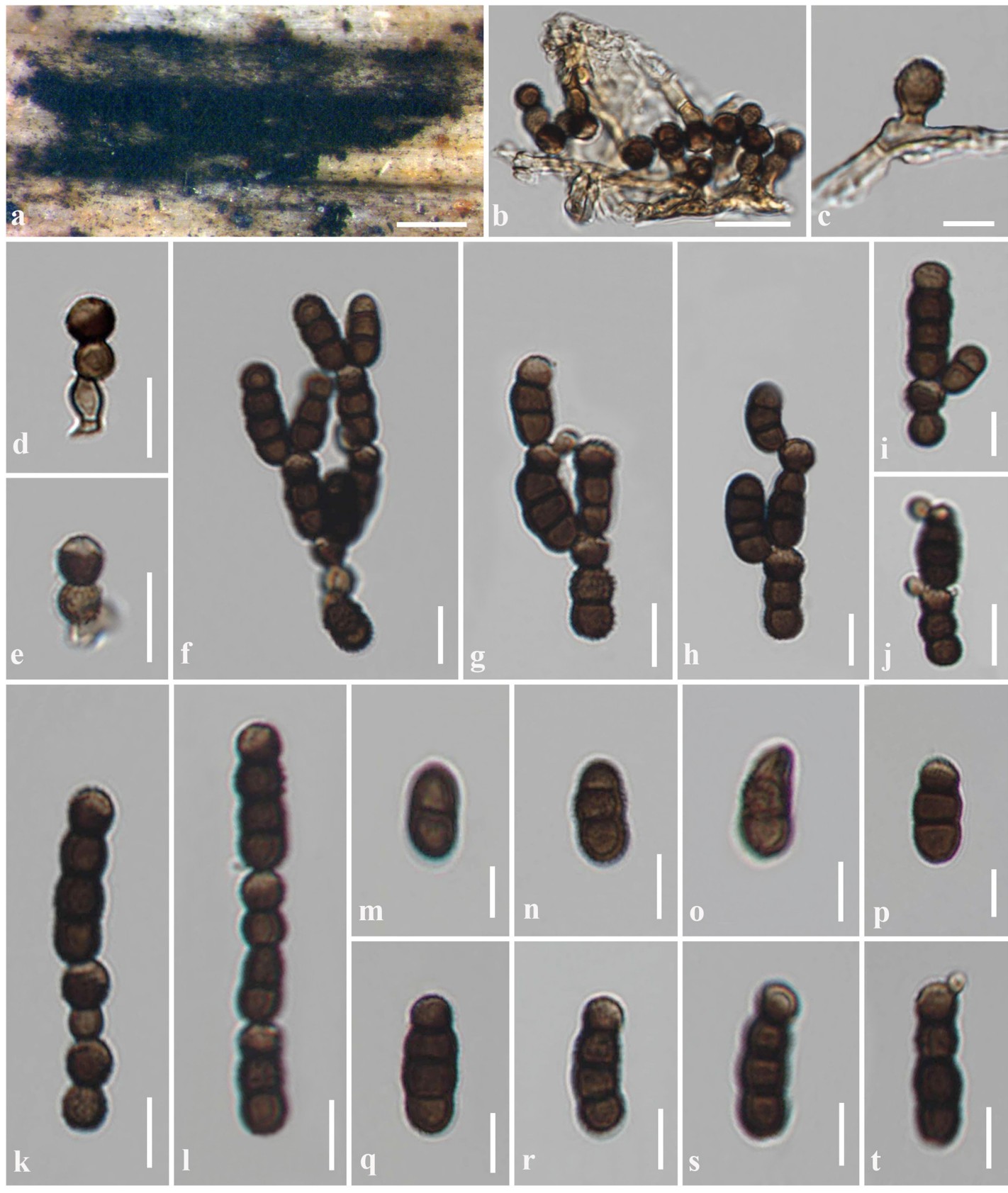

**Fig 3. *Torula hydei* (HKAS 97478, holotype).** a Colonies on dead branch of Chromolaena odorata. b–e Conidiophores with conidiogenous cell. f–j Budding on conidia. k, l Conidia in chain. m–t Conidia. Scale bars: a = 100 µm, b, k–l = 5 µm, c, f–j, q–t = 2 µm, d, e, m–p = 1 µm.

with a pale brown apex and the fertile cells in the conidial chain, where branching occurs, are darker brown than other cells [8]. The conidiogenous cells of *T. fici* are slightly larger than *T. hydei* and frequently clavate (*T. fici*, (5–)6(–8) × 5(–7) µm versus 3–5.5 × 4.3–5 µm, *T. hydei*), whereas, *T. hydei* has doliiform to ellipsoid conidiogenous cells [8]. We also note distinct nucleotide base pair differences between *T. hydei* and *T. fici* (CBS 595.96, type strain) across the ITS gene region (8/479 bp, 1.7% difference, no gap) and TEF1-α gene region analysed (43/ 760 bp, 5.7% difference, no gap). Based on distinct morphological characteristics and phylogenetic support, *T. hydei* is introduced as a new species in this study.

## Discussion

Taxonomic characterizations of taxa in Torulaceae have been well-studied since Crous et al. [8] who re-classified *Torula* and *Dendryphion* in Torulaceae (Pleosporales, Dothideomycetes) based on phylogenetic analyses of LSU sequence data. Subsequent authors introduced new genera and species in this family based on multigene phylogenetic analyses coupled with morphological characteristics (see Table 3) [10,18, 9, 5, 6, 7, 20]. Recently, there are more than 520 epithets in the genus *Torula* and 85 epithets in *Dendryphion* listed in Index Fungorum [26]. However, most of the described species lack DNA sequence data to verify their phylogenetic placement and affinities with other related fungi. Nevertheless, many species previously described as *Torula* and *Dendryphion* have also been synonymized to many genera in Sordariomycetes [26]. Taxa in these genera need to be clarified based on molecular data.

**Table 3. Synopsis of morphological features of the genera in Torulaceae.**

| Genus | Morphological features | | | Reference |
|---|---|---|---|---|
| | Conidia | Conidiophores | Conidiogenous cells | |
| *Dendryphion* | Acropleurogenous, catenate or solitary, simple or branched, cylindrical to obclavate, or cheiroid, pale to mid brown or olivaceous brown, multi-septate, smooth or verrucose | Macronematous, mononematous, branched at the apex, brown to black, smooth or with verruculose at the upper part, with paler branches | Mono- or polytretic, integrated, terminal and intercalary on branches, sympodial, clavate, cylindrical or doliiform, cicatrized, with large and dark scars. | [6,7] |
| *Neotorula* | Acrogenous, in chains, clavate to subcylindrical, septate, dark bands at the septa, pale green when young, brown when mature, verruculose | Macronematous, mononematous, cylindrical, 3–6-septate, with one or several short branches near the apex, smooth, dark brown, paler towards the apex | Tretic, with a distinct pore, integrated, terminal, pale brown or subhyaline, doliiform or lageniform | [6] |
| *Rostriconidium* | Solitary, pyriform to rostrate, dark brown to black, with a thick, black truncate scar at the base and pale pigment cell above the scar, narrowly cylindrical and obtuse at the apex | Macronematous, mononematous, single or caespitose, septate, smooth, brown or dark brown, unbranched, thick-walled, cylindrical, arising from a stromatic base. | Monotretic or polytretic, integrated, terminal, cylindrical, dark brown | [7] |
| *Rutola* | Acrogenous, simple to branched chains, phragmosporous, brown, verruculose, aseptate to multi-septate, fragmenting into segments | Micronematous, appressed to substrate, branched, septate, pale brown | Monoblastic, integrated, terminal or intercalary, pale brown | [10] |
| *Sporidesmioides* | Acrogenous, solitary, pyriform to rostrate, ampulliform to obclavate, truncate at the base, septate, brown to dark brown, with paler at the upper end cells, smooth or verruculose to echinulate | Macronematous, mononematous, scattered, unbranched, straight to curved, sometimes percurrently proliferating | Polyblastic, integrated, indeterminate or percurrent, terminal, sometimes intercalary sympodial, dark and prominent, cylindrical or doliiform. | [9] |
| *Torula* | Acrogenous, in branched chains, subcylindrical to cylindrical, brown, constricted at septa, smooth to verrucose, conidial cells subglobose | Micronematous, reduced to conidiogenous cells, or with a brown supporting cell | Mono- to polyblastic, solitary on mycelium, doliiform to ellipsoid or clavate, cupulate, brown, smooth to verruculose, | [8,5,6] |

*Torula* and *Dendryphion* have a wide host range in various habitats and are commonly found as saprobes in both terrestrial and aquatic habitats in temperate to tropical regions [10,3,59,18,9, 5,6,7,20]. It is interesting to note that many *Torula* species have been found to be associated with the host family Asteraceae [59,5]. In this study, our new strains were collected from Asteraceae and Li et al. [5] also reported two novel *Torula* species, *T. chromolaenae* and *T. mackenziei* from Asteraceae, indicating that Asteraceae harbors a diversity of these taxa. *Dendryphion hydei* was also collected from *Bidens pilosa* (Asteraceae) and is the first record from northern Thailand.

## Acknowledgments

The authors acknowledge the Biology Experimental Center, Germplasm Bank of Wild Species, Kunming Institute of Botany, Chinese Academy of Sciences to provide molecular laboratory facilities for molecular work. Itthayakorn Promputtha grateful to thank Chiang Mai University for partially support of this research work. Rajesh Jeewon would like to thank Mae Fah Luang University for giving him the opportunity as a visiting professor to the Center of Excellence in Fungal Research and University of Mauritius for research support. Mingkwan Doilom would like to thank the 5th batch of Postdoctoral Orientation Training Personnel in Yunnan Province and the 64th batch of China Postdoctoral Science Foundation for research support. We thanks to Emeritus Prof. Kevin D. Hyde, Dr. Shaun Pennycook, Dr. Dhanushka Wanasinghe, Hong-Bo Jiang, Dr. Zonglong Luo for their available suggestions and help.

## Author Contributions

**Conceptualization:** Junfu Li, Rajesh Jeewon, Rungtiwa Phookamsak.

**Data curation:** Junfu Li.

**Formal analysis:** Junfu Li, Rajesh Jeewon, Rungtiwa Phookamsak.

**Funding acquisition:** Rungtiwa Phookamsak, Itthayakorn Promputtha.

**Investigation:** Junfu Li, Rajesh Jeewon, Mingkwan Doilom, Rungtiwa Phookamsak.

**Methodology:** Junfu Li, Rungtiwa Phookamsak.

**Project administration:** Rungtiwa Phookamsak.

**Supervision:** Rajesh Jeewon, Peter E. Mortimer, Itthayakorn Promputtha.

**Writing – original draft:** Junfu Li, Mingkwan Doilom, Rungtiwa Phookamsak.

**Writing – review & editing:** Rajesh Jeewon, Peter E. Mortimer, Itthayakorn Promputtha.

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
