## [Decision Letter · Decision Letter 0]

31 Oct 2019

PONE-D-19-27091

Multigene phylogeny and taxonomy of Torula hydei and Dendryphion hydei spp. nov. from herbaceous litter in northern Thailand

PLOS ONE

Dear Dr. PROMPUTTHA,

Thank you for submitting your manuscript to PLOS ONE. After careful consideration, we feel that it has merit but does not fully meet PLOS ONE’s publication criteria as it currently stands. Therefore, we invite you to submit a revised version of the manuscript that addresses the points raised during the review process.

We would appreciate receiving your revised manuscript by Dec 15 2019 11:59PM. To enhance the reproducibility of your results, we recommend that if applicable you deposit your laboratory protocols in protocols.io, where a protocol can be assigned its own identifier (DOI) such that it can be cited independently in the future. For instructions see: http://journals.plos.org/plosone/s/submission-guidelines#loc-laboratory-protocols

We look forward to receiving your revised manuscript.

Kind regards,

Tzen-Yuh Chiang

Academic Editor

PLOS ONE

Journal Requirements:

2. Please take this opportunity to be sure you have met all of our guidelines for new species. When publishing papers that describe a new fungal taxon name, PLOS aims to comply with the requirements of the International Code of Nomenclature for algae, fungi, and plants (ICN). The following guidelines for publication in an online-only journal have been agreed such that any scientific fungal name published by us is considered effectively published under the rules of the Code. Please note that these guidelines differ from those for zoological nomenclature.

Effective January 2012, "the description or diagnosis required for valid publication of the name of a new taxon" can be in either Latin or English. This does not affect the requirements for scientific names, which are still to be Latin.

Also effective January 2012, the electronic PDF represents a published work according to the ICN for algae, fungi, and plants. Therefore the new names contained in the electronic publication of a PLOS ONE article are effectively published under that Code from the electronic edition alone, so there is no longer any need to provide printed copies.

For proper registration of the new taxon, we require two specific statements to be included in your manuscript.

In the Results section, the globally unique identifier (GUID), currently in the form of a Life Science Identifier (LSID), should be listed under the new species name, for example:

Hymenogaster huthii. Stielow et al. 2010, sp. nov. [urn:lsid:indexfungorum.org:names:518624]

You will need to contact either Mycobank or Index Fungorum to obtain the GUID (LSID).

In the Methods section, include a sub-section called "Nomenclature" using the following wording (this example is for taxon names submitted to MycoBank; please substitute appropriately if you have submitted to Index Fungorum and use the prefix http://www.indexfungorum.org/Names/NamesRecord.asp?RecordID= ):

*The electronic version of this article in Portable Document Format (PDF) in a work with an ISSN or ISBN will represent a published work according to the International Code of Nomenclature for algae, fungi, and plants, and hence the new names contained in the electronic publication of a PLOS ONE article are effectively published under that Code from the electronic edition alone, so there is no longer any need to provide printed copies*.

*In addition, new names contained in this work have been submitted to MycoBank from where they will be made available to the Global Names Index. The unique MycoBank number can be resolved and the associated information viewed through any standard web browser by appending the MycoBank number contained in this publication to the prefix http://www.mycobank.org/MB/. The online version of this work is archived and available from the following digital repositories: [INSERT NAMES OF DIGITAL REPOSITORIES WHERE ACCEPTED MANUSCRIPT WILL BE SUBMITTED (PubMed Central, LOCKSS etc)]*.

All PLOS ONE articles are deposited in PubMed Central and LOCKSS. If your institute, or those of your co-authors, has its own repository, we recommend that you also deposit the published online article there and include the name in your article.

A complete explanation of our guidelines for publishing new species can be found on our website: http://www.plosone.org/static/guidelines#fungal

5. Thank you for stating the following in the Acknowledgements Section of your manuscript:

"The authors are grateful to the Mushroom Research Foundation, Chiang Mai, Thailand and

366 Key Research Program of Frontier Sciences of the Chinese Academy of Sciences (grant no.

367 QYZDY-SSW-SMC014) for supporting this research. We also acknowledge that Biology

368 Experimental Center, Germplasm Bank of Wild Species, Kunming Institute of Botany, Chinese

369 Academy of Sciences provide molecular laboratory facilities for molecular work. Rungtiwa

370 Phookamsak thanks CAS President’s International Fellowship Initiative (PIFI) for young staff

371 (grant no. Y9215811Q1) and National Science Foundation of China (NSFC) project code

372 31850410489 (grant no. Y81I982211) for financial support. Itthayakorn Promputtha grateful

to thank Chiang Mai University for partially support 373 of this research work. Rajesh Jeewon

374 would like to thank Mae Fah Luang University for giving him the opportunity as a visiting

375 professor to the Center of Excellence in Fungal Research and University of Mauritius for research

376 support. Peter E Mortimer would like to thank the National Science Foundation of China and

377 the South East Asian Biodiversity resources Institute, Chinese Academy of Sciences, for

378 financial support under the following grants: 41761144055, 41771063, Y4ZK111B01.

379 Mingkwan Doilom would like to thank the 5th batch of Postdoctoral Orientation Training

380 Personnel in Yunnan Province and the 64th batch of China Postdoctoral Science Foundation.

381 Jun-Fu Li thanks to Emeritus Prof. Kevin D. Hyde, Dr. Shaun Pennycook, Dr. Dhanushka

382 Wanasinghe, Hong-Bo Jiang, Dr. Zonglong Luo for their available suggestions and help."

We note that you have provided funding information that is not currently declared in your Funding Statement. However, funding information should not appear in the Acknowledgements section or other areas of your manuscript. We will only publish funding information present in the Funding Statement section of the online submission form.

 "No".

Reviewers' comments:

Reviewer's Responses to Questions

**Comments to the Author**

1. Is the manuscript technically sound, and do the data support the conclusions?

Reviewer #1: Yes

Reviewer #2: Yes

2. Has the statistical analysis been performed appropriately and rigorously? 

Reviewer #1: Yes

Reviewer #2: N/A

3. Have the authors made all data underlying the findings in their manuscript fully available?

Reviewer #1: Yes

Reviewer #2: Yes

4. Is the manuscript presented in an intelligible fashion and written in standard English?

Reviewer #1: Yes

Reviewer #2: Yes

5. Review Comments to the Author

Reviewer #1: Dear Junfu Li and co-authors,

Please see the attachment.

I have included several comments in the attached manuscript. Basically, this is a well written manuscript, however, i am not sure the content is strong enough for the Plos ONE.

Please remove the unnecessary paragraph from the introduction, and please improve your discussion. Most parts related to the Discussion should move to the notes part.

Please improve your phylogeny by including Rutola. Please refer "The Genera of Fungi – G5: Arthrinium, Ceratosphaeria, Dimerosporiopsis, Hormodochis, Lecanostictopsis, Lembosina, Neomelanconium, Phragmotrichum, Pseudomelanconium, Rutola, and Trullula"

Reviewer #2: Note: The following minor corrections may please be attended -

Page 8, Lines 26-27: Abstract: Delete the first sentence. Start the paragraph from second sentence – During our studies....

Pages 8-9, Lines: 44-48: Replace the sentences with this: Species in Torulaceae are known only by their asexual morphs which are characterized as follows:Superficial, effuse, greyish brown to black, powdery colonies; micro- or macronematous conidiophores, with or without apical branches; doliiform to ellipsoid or clavate, brown, smooth to verruculose, mono- to polyblastic conidiogenous cells which often remaining cupulate; subcylindrical, phragmosporous, acrogenous, brown, dry, smooth to verrucose conidia characteristically produced in branched chains

Page 9, L.50: accepted Dendryphion Wallr., besides Torula, within Torulaceae in Pleosporales.

Page 9, L.54: relationships of the taxa within the family ...

Page 11, L.94: northern Thailand. Among collected samples, Dendryphion hydei is recovered as another new .....

Page 11, L.96: updated phylogenetic tree with our new taxa for the family Torulaceae is provided in this ...

Page 12, L.113: Single conidium isolation ..

Page 24, L. 287: semi-macronematous, mononematous, ....

Page 24, L. 289: subglobose, arising from prostrate hyphae

Page 25, L. 315: al. [8] who re-classified Torula and ....

Page 25, L. 316: ......... Subsequent authors introduced new genera ....

Page 25, L. 320: Fungorum [23], but most of the described species lack DNA .....

6. PLOS authors have the option to publish the peer review history of their article (what does this mean?). If published, this will include your full peer review and any attached files.

Reviewer #1: No

Reviewer #2: No

---

## [Author Response · Author response to Decision Letter 0]

20 Dec 2019

Dear reviewers,

Thank you very much for your valuable comments and suggestions. We have revised our manuscript following your comments.

The phylogenetic tree is updated by including the genus Rotula and some other missing strains. Those requirements from the reviewers are provided and highlighted as yellow in the revised manuscript with track changes.

We will appreciate if this article is accepted and published in PLOS ONE. We are willing to provided more information if you have more requirement. Please feel free to contact us if you have more question. 

We are looking forward to hearing from you.

Yours sincerely,

Jun-Fu Li/Rungtiwa Phookamsak/ Itthayakorn Promputtha

---

## [Decision Letter · Decision Letter 1]

8 Jan 2020

Multigene phylogeny and taxonomy of Dendryphion hydei and Torula hydei spp. nov. from herbaceous litter in northern Thailand

PONE-D-19-27091R1

Dear Dr. PROMPUTTHA,

We are pleased to inform you that your manuscript has been judged scientifically suitable for publication and will be formally accepted for publication once it complies with all outstanding technical requirements.

With kind regards,

Tzen-Yuh Chiang

Academic Editor

PLOS ONE

Additional Editor Comments (optional):

Reviewers' comments:

Reviewer's Responses to Questions

**Comments to the Author**

1. If the authors have adequately addressed your comments raised in a previous round of review and you feel that this manuscript is now acceptable for publication, you may indicate that here to bypass the “Comments to the Author” section, enter your conflict of interest statement in the “Confidential to Editor” section, and submit your "Accept" recommendation.

Reviewer #1: All comments have been addressed

Reviewer #2: All comments have been addressed

2. Is the manuscript technically sound, and do the data support the conclusions?

Reviewer #1: Yes

Reviewer #2: Yes

3. Has the statistical analysis been performed appropriately and rigorously? 

Reviewer #1: Yes

Reviewer #2: Yes

4. Have the authors made all data underlying the findings in their manuscript fully available?

Reviewer #1: Yes

Reviewer #2: Yes

5. Is the manuscript presented in an intelligible fashion and written in standard English?

Reviewer #1: Yes

Reviewer #2: Yes

6. Review Comments to the Author

Reviewer #1: Dear Author

Thanks for addressing the comments raised during previous round.

I am happy with the MS and now it ready for publish

Reviewer #2: The authors have incorporated all corrections and suggestions made by the reviewers' and the manuscript is in very good shape. The paper is now recommended for publication.

7. PLOS authors have the option to publish the peer review history of their article (what does this mean?). If published, this will include your full peer review and any attached files.

Reviewer #1: No

Reviewer #2: No

---

## [Editor Report · Acceptance letter]

14 Jan 2020

PONE-D-19-27091R1 

Multigene phylogeny and taxonomy of *Dendryphion hydei* and *Torula hydei* spp. nov. from herbaceous litter in northern Thailand 

Dear Dr. PROMPUTTHA:

I am pleased to inform you that your manuscript has been deemed suitable for publication in PLOS ONE. Congratulations! Your manuscript is now with our production department. 

With kind regards,

on behalf of

Dr. Tzen-Yuh Chiang 

Academic Editor

PLOS ONE